# FastKAN-DDD: A novel fast Kolmogorov-Arnold network-based approach for driver drowsiness detection optimized for TinyML deployment

**Siham Essahraui**[1], **Ismail Lamaakal**[1], **Yassine Maleh**[2]*, **Khalid El Makkaoui**[1], **Mouncef Filali Bouami**[1], **Ibrahim Ouahbi**[1], **Hela Elmannai**[3], **Ahmed A. Abd El-Latif**[4,5]

**1** Multidisciplinary Faculty of Nador, Mohammed Premier University, Oujda, Morocco, **2** Laboratory LaSTI, ENSAK, Sultan Moulay Slimane University, Khouribga, Morocco, **3** Department of Information Technology, College of Computer and Information Sciences, Princess Nourah bint Abdulrahman University, Riyadh, Saudi Arabia, **4** EIAS Data Science Lab, College of Computer and Information Sciences, and Center of Excellence in Quantum and Intelligent Computing, Prince Sultan University, Riyadh, Saudi Arabia, **5** Department of Mathematics and Computer Science, Faculty of Science, Menoufia University, Shebin El-Koom 32511, Egypt

☉ These authors contributed equally to this work.

* yassine.maleh@ieee.org

**Data availability statement:** Data sources are included in the paper.

## Abstract

Driver drowsiness is a leading cause of traffic accidents and fatalities, highlighting the urgent need for intelligent systems capable of real-time fatigue detection. Although recent advancements in machine learning (ML) and deep learning (DL) have significantly improved detection accuracy, most existing models are computationally demanding and not well-suited for deployment in resource-limited environments such as microcontrollers. While the emerging domain of TinyML presents promising avenues for such applications, there remains a substantial gap in the development of lightweight, interpretable, and high-performance models specifically tailored for embedded automotive systems. This paper introduces FastKAN-DDD, an innovative driver drowsiness detection model grounded in the Fast Kolmogorov-Arnold Network (FastKAN) architecture. The model incorporates learnable nonlinear activation functions based on radial basis functions (RBFs), facilitating efficient function approximation with a minimal number of parameters. To enhance suitability for TinyML deployment, the model is further optimized through post-training quantization techniques, including dynamic range, float-16, and weight-only quantization. Comprehensive experiments were conducted using the UTA-RLDD dataset—a real-world benchmark for driver drowsiness detection—evaluating the model across various input resolutions and quantization schemes. The FastKAN-DDD model achieved a test accuracy of 99.94%, with inference latency as low as 0.04 ms and a total memory footprint of merely 35 KB, rendering it exceptionally well-suited for real-time inference on microcontroller-based systems. Comparative evaluations further confirm that FastKAN surpasses several state-of-the-art TinyML models in terms of

**Funding:** Princess Nourah bint Abdulrahman University Researchers Supporting Project number (PNURSP2025R747), Princess Nourah bint Abdulrahman University, Riyadh, Saudi Arabia. The funders had a role in the decision to publish and the preparation of the manuscript.

**Competing interests:** NO authors have competing interests.

accuracy, computational efficiency, and model compactness. Our code's are publicly available at: https://github.com/sihamess/driver_drowsiness_detection_TinyML.

# 1 Introduction

Driver drowsiness remains a major cause of road accidents and traffic-related fatalities globally, contributing significantly to incidents characterized by delayed response times, impaired decision-making, and compromised vehicle control [1]. In response to the increasing integration of intelligent safety features in modern vehicles, considerable research attention has been directed toward the development of effective Driver Drowsiness Detection (DDD) systems capable of real-time monitoring to enhance driver and passenger safety. Existing DDD methodologies primarily rely on physiological signals (e.g., EEG, ECG, EDA) [2,3], behavioral indicators (e.g., yawning, eye closure, head movements) [4–6], or vehicular dynamics (e.g., steering patterns) [7,8]. While these approaches have demonstrated potential, their practical application—particularly in low-cost, real-time embedded systems—continues to pose significant technical challenges.

Recent progress in ML and DL [9–12] has led to improved accuracy and responsiveness in DDD systems, especially those leveraging computer vision [13]. Nevertheless, many of these models are computationally intensive and unsuitable for deployment in resource-constrained environments such as microcontrollers or edge devices, where constraints on power consumption, memory, and latency are critical. Although TinyML—a paradigm aimed at enabling ML execution on ultra-low-power hardware—has garnered increasing interest, existing DDD models still grapple with the trade-offs between performance, interpretability, and deployability [14,15].

To overcome these limitations, the present study introduces a novel model based on the Fast Kolmogorov-Arnold Network (FastKAN-DDD) [16], specifically optimized for real-time driver drowsiness detection in TinyML environments. This model draws upon the Kolmogorov–Arnold representation theorem by substituting conventional weight matrices with learnable univariate functions, thereby enhancing model expressiveness, compactness, and interpretability. The integration of radial basis functions and a streamlined network architecture facilitates accurate classification of drowsy states while maintaining minimal memory usage and inference latency, rendering the approach particularly well-suited to embedded automotive systems.

The primary contributions of this research are summarized as follows:

- A novel FastKAN-based architecture for DDD is proposed, offering a balance between accuracy, model compactness, and interpretability.
- Comprehensive experimental evaluations are conducted using the UTA-RLDD dataset, examining model performance across varying input resolutions and quantization schemes tailored for edge deployment.
- The proposed model is benchmarked against current state-of-the-art TinyML-based DDD systems, demonstrating superior performance in terms of inference speed, classification accuracy, and practical deployability.

The remainder of this paper is organized as follows: Sect 2 reviews the relevant literature on ML, DL, and TinyML approaches to DDD. Sect 3 outlines the proposed methodology, including dataset characteristics, architectural design, and quantization methods. Sect 4 details the experimental configuration, evaluation procedures. Sect 5 discusses the obtained

results, compares the findings with prior studies, and addresses deployment considerations. Finally, Sect 6 concludes the study and suggests avenues for future work.

## 2 Literature review

This section offers a thorough review of recent research focused on enhancing DDD using AI. It begins by examining both traditional ML and advanced DL approaches that analyze physiological and visual indicators of fatigue. The section then highlights the growing importance of TinyML in powering real-time, energy-efficient DDD systems designed for embedded environments.

### 2.1 Machine learning and deep learning in driver drowsiness detection

Driver tiredness identification has been extensively studied utilizing diverse machine learning methodologies to improve road safety. Recent studies have investigated facial analysis with machine learning, deep learning, and hybrid frameworks to enhance accuracy and reliability in diverse driving conditions.

A real-time driver drowsiness detection system was developed using machine learning and deep learning [17] models based on face analysis [9]. The research evaluated K-Nearest Neighbors (KNN), Support Vector Machines (SVM), Convolutional Neural Networks (CNNs), YOLOv5, YOLOv8, and Faster R-CNN using the NTHUDDD, YawDD, and UTA-RLDD datasets. YOLOv5 and YOLOv8 attained 100% precision and recall, while KNN obtained 98.89% accuracy on UTA-RLDD. The system effectively monitors and analyzes driver facial expressions for drowsiness detection with high accuracy in real-time driver monitoring applications. An advanced driver-fatigue detection system integrated deep CNNs with emotional state monitoring to improve detection accuracy under varying illumination conditions [18]. The system underwent evaluation using the Yawn and YawDDR datasets, attaining an accuracy of 95.3% on YawDDR. The model utilizes dynamic brightness adaption methods, guaranteeing real-time monitoring in challenging environmental conditions such as tunnels and varying light exposure. Further extending CNN-based fatigue detection, a real-time system focused on yawning and eye state monitoring to assess driver drowsiness [19]. The system was trained on video datasets with diverse lighting conditions and face angles, employing Haar cascade classifiers for facial extraction and deep learning models for fatigue detection. The model attained a testing accuracy of 96.54%, indicating exceptional efficacy in real-time drowsiness detection. To address the limitations of conventional fatigue detection models when drivers wear sunglasses, an alternative approach employed YOLOv8 with transfer learning and infrared imagery [20]. Conventional visual fatigue detection models encounter difficulties with blocked facial features; however, this method utilizes infrared cameras to identify eye closure, yawning, and prolonged drowsiness. The system attained 98% detection accuracy by using a threshold-based classification algorithm that evaluates closed-eye frame ratios. Beyond driver fatigue, an intelligent transportation monitoring system was introduced to integrate real-time analysis of both driver behavior and surrounding traffic environments [21]. The system employs YOLOv9 for traffic object detection, enhancing the accuracy of vehicle and pedestrian recognition while concurrently observing driver behavior, including fatigue and aggressive driving patterns. SafeSmartDrive attained 83.1% accuracy in traffic detection and presents a multi-layer AI-driven framework for enhancing road safety.

Overall, conventional machine learning and deep learning pipelines for driver drowsiness detection report strong performance on curated benchmarks and controlled testbeds. End-to-end CNN or YOLO detectors, as well as transfer-learning variants—sometimes augmented with infrared sensing—achieve high accuracy and low false-alarm rates with real-time

throughput on GPU-class hardware. Nevertheless, several factors constrain their translation to in-vehicle deployment at scale: (i) substantial compute, memory, and energy demands that exceed typical embedded automotive budgets; (ii) sensitivity to illumination changes, head pose, camera placement, and occlusions (e.g., sunglasses, masks); (iii) dataset bias and distribution shift across drivers, vehicles, and sensors, which limit cross-dataset generalization; (iv) insufficient model calibration, uncertainty quantification, and interpretability, which impede failure analysis and human oversight; (v) privacy, security, and regulatory concerns associated with continuous in-cabin video capture; and (vi) a paucity of prospective, longitudinal, and multi-site validations under standardized protocols.

These limitations motivate the development of TinyML-oriented edge solutions, multimodal fusion (e.g., combining cameras with EEG/PPG or vehicle signals), domain adaptation and continual learning strategies, and rigorous cross-dataset evaluation to improve reliability, fairness, and real-world readiness.

## 2.2 TinyML for driver drowsiness detection

TinyML [22,23] is an emerging domain that enables the deployment of machine learning models on low-power, resource-limited devices, including microcontrollers and edge devices. This capability is especially beneficial for real-time applications such as driver drowsiness detection, where continuous monitoring and rapid inference are essential for road safety. Recent research has investigated TinyML-based systems for detecting driver sleepiness, utilizing various approaches and architectures.

Gwo-Ching Chang et al. [24] developed an enhanced YOLOv7-Tiny model for real-time driver fatigue monitoring focusing on eye state detection. The model was improved by structured pruning and architectural fine-tuning, resulting in a 97% decrease in size and a sixfold enhancement in processing performance, while preserving high accuracy, recall, and mean average precision (mAP). The improved model size decreased from 74.8 MB (YOLOv7) to 7.1 MB (YOLOv7-Tiny-Second-Prune). The model is still relatively large compared to typical TinyML implementations and may not be fully suited for deployment on ultra-low-power microcontrollers. Meanwhile, Norah N. Alajlan et al. [25] created a TinyML-driven system for detecting driver drowsiness, optimized for deployment on microcontrollers. The research employed five lightweight deep learning architectures (SqueezeNet, AlexNet, CNN, MobileNet-V2, and MobileNet-V3) and utilized quantization methods (QAT, FIQ, DRQ) to reduce model sizes. The smallest deployed model was CNN (0.05 MB utilizing DRQ), succeeded by SqueezeNet (0.141 MB), AlexNet (0.58 MB), MobileNet-V3 (1.16 MB), and MobileNet-V2 (1.55 MB). The highest accuracy achieved was 99.64% (MobileNet-V2 using DRQ). Ha-Trung Nguyen et al. [26] developed a wearable EEG-based system for detecting driver drowsiness, utilizing behind-the-ear (BTE) EEG sensors and Tiny Neural Networks for on-device classification. The research assessed Multilayer Perceptron (MLP) and CNN models, optimized for implementation on a microcontroller (MCU) with TensorFlow Lite for Microcontrollers (TFLM). The system was built on an nRF52840 microcontroller, featuring a 32-bit ARM Cortex-M4F CPU, 256 KB RAM, and 1 MB flash memory. The CNN-based model achieved high accuracy for drowsiness detection while significantly reducing the model size. Sebastian Frey et al. [27] created a wearable device for drowsiness detection utilizing EEG and PPG, designed for real-time inference at the edge. The system operates on BioGAP, an ultra-low-power acquisition and processing platform that incorporates the GAP9 System-on-Chip. A lightweight CNN model (21.6K parameters) was deployed on GAP9, achieving 91.1% accuracy in detecting drowsy states—a 6% improvement over single-modality EEG or PPG models. The optimized model size was 21.1 KB, facilitating

real-time processing with an inference time of 10.17 ms and an energy usage of just 0.36 mJ per inference, allowing continuous operation for 14 hours on a small 75 mAh battery.

Table 1 provides a comprehensive overview of current research on DDD employing TinyML techniques. It outlines the core objectives of each study, the methodological frameworks utilized, and the corresponding performance outcomes, thereby presenting a consolidated perspective on recent advancements in this field.

## 3 Methodology

The primary goal of this research is to design an accurate, interpretable, and computationally efficient DDD system tailored for deployment on edge devices with limited resources. By utilizing the recently introduced KANs, specifically their accelerated version, FastKAN, this work seeks to address the constraints of conventional deep learning models concerning model size, interpretability, and efficiency in real-time inference.

This section outlines the methodology adopted for the development and assessment of the FastKAN-based DDD system. It commences with a description of the dataset utilized, followed by the preprocessing procedures. The core architecture, grounded in KANs, is subsequently introduced, encompassing both the theoretical underpinnings and the performance enhancements foundation in the FastKAN variant. The following subsections elaborate on the model architecture, training setup, and performance metrics. Finally, post-training quantization methods are described to optimize the model for TinyML platforms with limited resources, concluding with the model conversion process required for embedded system integration.

### 3.1 Dataset description

The University of Texas at Arlington Real-Life Drowsiness Dataset (UTA-RLDD) [28] is a robust and comprehensive dataset designed for multi-stage drowsiness detection, encompassing both pronounced and subtle microexpressions indicative of fatigue. It comprises approximately 30 hours of RGB video footage collected from 60 healthy participants, with each individual contributing one video per drowsiness category—alert, low vigilance, and drowsy—resulting in a total of 180 videos. The dataset was recorded in real-world environments using participants' personal mobile phones or webcams, thereby ensuring natural variability in lighting conditions, camera angles, and backgrounds. To enhance realism, participants recorded their videos under conditions that simulate in-vehicle settings, aligning with real-life fatigue detection scenarios. The dataset has been curated and utilized due to its demonstrated effectiveness in our prior research [9]. A summary of its key attributes is presented in Table 2.

**Table 1**. **TinyML-compatible systems for DDD.**

| Ref. | Year | Aim | Method(s) | Performance |
|---|---|---|---|---|
| [25] | 2023 | TinyML-based drowsiness detection for microcontrollers | DRQ/QAT on CNN, SqueezeNet, AlexNet, MobileNet-V2/V3 | 99.64% (MobileNet-V2 with DRQ) |
| [26] | 2023 | Wearable EEG-based embedded drowsiness detection | Tiny MLP and CNN | High precision, real-time portable EEG classification |
| [24] | 2024 | Real-time eye-state detection via lightweight YOLOv7-Tiny variant | YOLOv7-Tiny | mAP preserved, 6× speedup, 97% size reduction |
| [27] | 2024 | Real-time drowsiness detection using EEG+PPG on BioGAP SoC | Lightweight CNN | 91.1% accuracy, 10.17 ms inference, 0.36 mJ per inference |

**Table 2**. **Summary of the UTA-RLDD dataset used for driver drowsiness detection.**

| Attribute | Details |
|---|---|
| Number of participants | 60 |
| Total video duration | Approximately 30 hours |
| Number of videos | 180 |
| Drowsiness categories | Alert, Low Vigilance, Drowsy |
| Frame rate | Less than 30 fps |
| Recording devices | Personal mobile phones and webcams |
| Recording environments | Real-world settings (e.g., home, university) |

## 3.2 Data preprocessing

To prepare the video datasets for model training, we systematically converted each video into individual frames. This step allowed us to categorize each frame into one of two classes: 'awake' (8,750 images) or 'drowsy' (8,089 images). This meticulous classification process was crucial for creating a well-structured labeled dataset, forming the foundation for both effective model training and thorough evaluation. Once labeled, the frames were carefully divided into separate subsets for training, validation, and testing, ensuring a balanced dataset that facilitates robust performance assessment and reliable model validation.

Sampling, quality control, and splits: Each RLDD video (*Alert*, *Low-Vigilance*, *Drowsy* per subject) was uniformly sampled at 1 frame per second; we then removed the first and last 10% of frames to avoid state transitions. Frames were retained only if a face was detected and passed basic blur/illumination checks; to reduce redundancy we kept a frame only when its SSIM to the last retained frame was <0.98. We enforced per-video quotas to maintain class and subject balance. For binary classification we mapped Alert → Awake and Low-Vigilance + Drowsy → Drowsy. After filtering we obtained 20,205 labeled frames; we performed a subject-disjoint split with 48 subjects for train/val (85/15 within those subjects) and 12 subjects for test, yielding 16,839 train+val and 3,366 test frames (test distribution: 1,722 Awake / 1,644 Drowsy). This protocol limits temporal autocorrelation, prevents identity leakage, and preserves variation in pose and illumination.

To reduce redundancy without narrowing the data distribution, we uniformly subsampled each of the 180 RLDD videos at 1 fps with quality-control filtering and per-video quotas. This preserved variation across subjects, recording devices, lighting, and head pose while preventing any single clip from dominating. We then performed a subject-disjoint split (train/val on 48 subjects; test on 12 held-out subjects), so that evaluation measures generalization to unseen identities and capture conditions rather than memorization of near-duplicate frames. Consequently, the reduced training and test sets maintain the same diversity of conditions as the complete dataset at the temporal granularity used for learning, and the model trained on 16k samples achieves comparable performance on the held-out set, indicating robust generalization across the full dataset.

## 3.3 Kolmogorov-Arnold network model design

Kolmogorov-Arnold Networks (KANs) [29–31] are a novel neural network architecture inspired by the Kolmogorov–Arnold representation theorem. Unlike traditional Multi-Layer Perceptrons (MLPs), which use fixed activation functions on neurons, KANs introduce learnable activation functions on edges (weights). Instead of using linear weight matrices, KANs [32,33] replace each weight with a univariate function parameterized as a spline. This unique approach improves accuracy and interpretability in function approximation tasks.

### 3.3.1 Key equations of KAN.

- **KAN Representation Theorem–Based Approximation**

  By the Kolmogorov–Arnold theorem, any continuous function $f\colon \mathbb{R}^n \to \mathbb{R}$ can be written as

$$f(x) = \sum_{j=1}^{2n+1} \Phi_j\left(\sum_{i=1}^{n} \varphi_{j,i}(x_i)\right), \tag{1}$$

  where $\varphi_{j,i}$ act on individual inputs and $\Phi_j$ are outer functions.

- **KAN Layer Transformation**

  Given activations $x^{(l)} \in \mathbb{R}^{n_l}$, a KAN layer produces

$$x_j^{(l+1)} = \Phi_j^{(l)}\left(\sum_{i=1}^{n_l} \varphi_{j,i}^{(l)}\left(x_i^{(l)}\right)\right), \qquad j = 1, \dots, n_{l+1}. \tag{2}$$

  Here, the outer-sum index $j$ in (1) corresponds to the output unit (channel) index of layer $l+1$, and $i$ indexes inputs from layer $l$. In our implementation, $\Phi_j^{(l)}$ is the identity (absorbed into $\varphi_{j,i}^{(l)}$), yielding the simplified form used in Eq. (2):

$$x_j^{(l+1)} = \sum_{i=1}^{n_l} \varphi_{j,i}^{(l)}\left(x_i^{(l)}\right). \tag{3}$$

- General KAN Model Composition

  A deep KAN with $L$ layers is represented as:

$$KAN(x) = \left(\Phi_{L-1} \circ \Phi_{L-2} \circ \dots \circ \Phi_1 \circ \Phi_0\right)(x) \tag{4}$$

  This contrasts with the standard MLP formulation:

$$MLP(x) = \left(W_{L-1} \circ \sigma \circ W_{L-2} \circ \sigma \circ \dots \circ W_1 \circ \sigma \circ W_0\right)(x) \tag{5}$$

  where:
  - $W_i$ are linear weight matrices in MLPs.
  - $\sigma$ is a fixed activation function (e.g., ReLU, Sigmoid).
  - KANs replace $W_i$ with learnable functions $\Phi_i$, allowing for more flexibility.

**3.3.2 Advantages of KAN over MLP.** KANs provide superior function approximation, interpretability, and efficiency in cases where traditional MLPs struggle, making them a promising alternative in AI and scientific applications (see Fig 1).

**3.3.3 Fast KAN.** Fast KAN [16] is an improved version of Kolmogorov-Arnold Networks (KANs) that use Gaussian radial basis functions (RBFs) in place of conventional 3-order B-splines. This invention greatly enhances computing efficiency without sacrificing model accuracy. FastKAN enhances neural network design with the integration of Gaussian RBFs, making it particularly adept at managing high-speed data processing requirements.

Gaussian Radial Basis Functions (RBFs) are essential in function approximation, defined by their reliance exclusively on the distance from a central point, known as the radial distance. These functions are expressed mathematically as:

$$\phi(r) = e^{-\frac{r^2}{2h^2}} \tag{6}$$

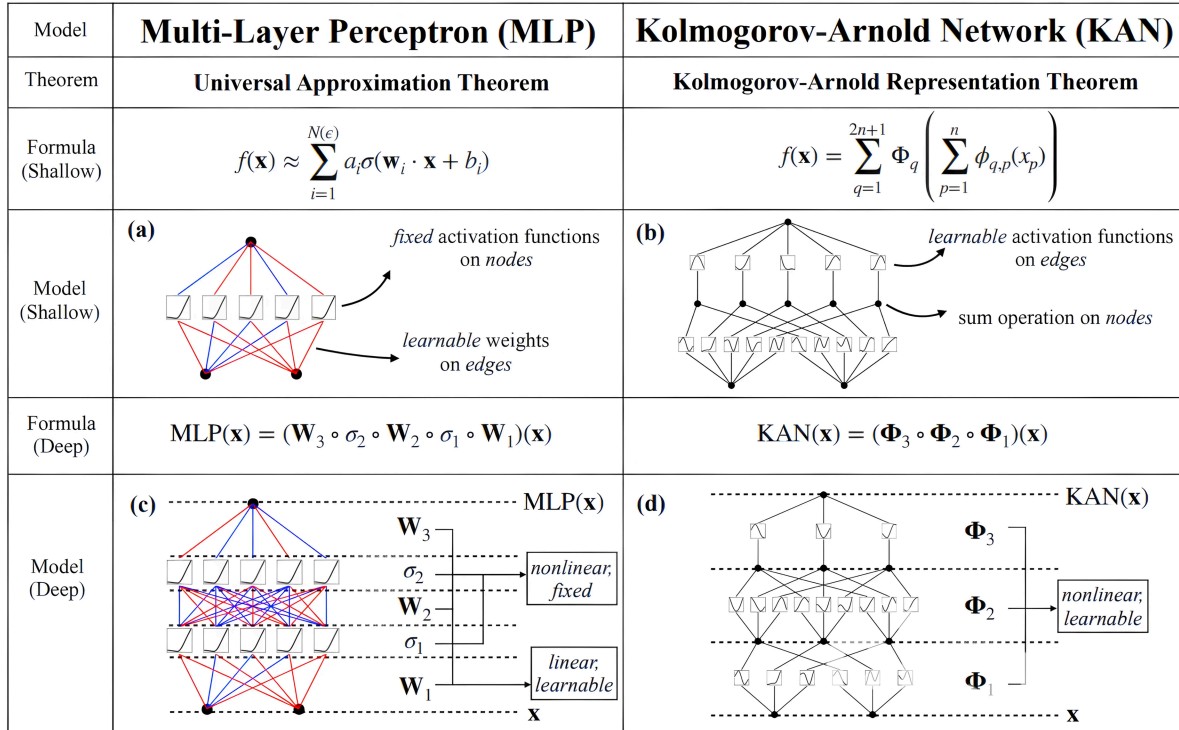

**Fig 1. Comparison between MLP and KAN as presented in [29].** The figure illustrates key differences between MLPs and KANs, including their respective theoretical foundations, mathematical formulations, and structural representations. Unlike MLPs, which use fixed activation functions on nodes and learnable weights on edges, KANs utilize learnable activation functions on edges and sum operations on nodes, leading to improved flexibility and interpretability.

Here, $r$ denotes the radial distance and $h$ is a parameter that controls the spread of the function. RBFs are favored for their efficacy in localized approximations, making them essential in several machine learning and pattern recognition applications.

FastKAN significantly enhances both the speed and precision compared to conventional KAN implementations. By substituting 3-order B-splines with Gaussian radial basis functions, FastKAN attains a forward computation speed approximately 3.33 times faster, hence reducing processing durations in conventional efficient KAN configurations. Moreover, accuracy assessments on benchmarks such as the MNIST dataset indicate that FastKAN either equals or exceeds the performance of traditional KANs, demonstrating its capacity to provide both rapid processing and great precision in modeling complex functions. FastKAN is an optimal selection for applications necessitating rapid and precise high-dimensional data processing.

Our proposed FastKAN model, incorporates a novel neural network architecture optimized for the efficient approximation of complicated, high-dimensional functions. The architecture consists of a series of custom FastKANLayers that employ RBFs for data processing, emphasizing both the local features and the broader trends within the dataset. Each layer is meticulously and finely tuned with specified characteristics, such as the number of grid points and their range, allowing accurate and adaptive management of input information (see Table 3). The layers employ a combination of Radial Basis and Linear (SiLU) activations (see Equation 7) to execute non-linear transformations, enabling robust feature extraction crucial for complex analytical tasks.

**Table 3**. Architecture and parameters of the FastKAN model.

| Layer Name | Type | Input Shape | Output Shape | Activation | Additional Info |
|---|---|---|---|---|---|
| Input | Input Layer | $W \times H$ | $W \times H$ | - | Raw Input |
| FastKANLayer 1 | Custom Layer (FastKANLayer) | $W \times H \to 5$ | 5 | Radial Basis + Linear (SiLU) | Uses RBF with 2 grid points, grid range: $[-1,1]$ |
| FastKANLayer 2 | Custom Layer (FastKANLayer) | $5 \to 2$ | 2 | Radial Basis + Linear (SiLU) | Uses RBF with 2 grid points, grid range: $[-1,1]$ |
| Output Layer | Dense | 2 | - | Softmax | Final classification layer |

**Training Parameters:**

- **Optimizer:** Adam (learning rate = $10^{-3}$)
- **Loss Function:** Sparse Categorical Crossentropy (from logits)
- **Batch Size:** 256
- **Epochs:** 20
- **Evaluation Metrics:** Sparse Categorical Accuracy
- **Gradient Optimization:** Applied via backpropagation with Adam optimizer
- **Weight Initialization:** Xavier/Glorot Initialization for Dense layers

SiLU is a smoothed version of ReLU, defined as:

$$\text{SiLU}(x) = x \cdot \sigma(x) \tag{7}$$

where $\sigma(x)$ is the sigmoid function:

$$\sigma(x) = \frac{1}{1 + e^{-x}} \tag{8}$$

SiLU improves gradient flow in deep networks and avoids the "dying ReLU" problem.

The last stage of this process occurs in the output layer, which is a Dense layer utilizing Softmax activation to categorize data into established classes, optimizing the model's efficacy for precise classification and pattern recognition tasks. The model is optimized using the Adam optimizer with a fixed learning rate of $10^{-3}$, and loss is handled by Sparse Categorical Crossentropy. Training for 20 epochs with a batch size of 256 ensures comprehensive learning without overfitting, supported by Xavier/Glorot Initialization for optimal weight configuration in dense layers. This strategic configuration facilitates rapid convergence and strong performance on novel, unseen data.

## 3.4 Model compression

This section explores the crucial process of model conversion, specifically adapting the proposed FastKAN model for deployment on resource-limited devices. Converting the model to TensorFlow Lite format is essential for operation within the constrained computing environments typical of edge devices. This transformation is achieved through three primary compression techniques: full integer quantization, float-16 quantization, and weight quantization.

**3.4.1 Post-training quantization techniques.** Quantization [34,35] is a technique used to optimize deep learning models by reducing the precision of numerical representations. Typically, deep learning frameworks use 32-bit floating-point (FP-32) tensors, which result in large model sizes and high computational demands. Quantization reduces these tensors to lower-bit representations, such as 8-bit integers (INT-8), enabling smaller model sizes and faster inference, particularly on resource-constrained hardware.

Mathematically, quantization is defined as:

$$Q(x) = \text{round}(x \times S + Z) \tag{9}$$

where:

- $x$ is the original floating-point value,
- $S$ is the scale factor,
- $Z$ is the zero-point,
- $Q(x)$ is the quantized integer value.

This conversion allows efficient computation while maintaining reasonable model accuracy.

**3.4.2 Dynamic range quantization.** This method converts model weights from FP-32 to INT-8 while keeping activations in floating-point format. Activations are quantized just before execution and dequantized afterward. This approach reduces model size but retains some floating-point computations.

$$Q_w = \text{round}\left(\frac{W}{S_w}\right) \tag{10}$$

where $W$ represents the model weight, and $S_w$ is the scale factor.

**3.4.3 Full integer quantization.** Full integer quantization converts both weights and activations to INT-8, enabling integer-only computations. It requires a representative dataset to determine dynamic ranges. This method ensures the highest inference speed on integer-only hardware.

$$Q_a = \text{round}\left(\frac{A}{S_a}\right) \tag{11}$$

where $A$ is the activation value, and $S_a$ is the activation scale factor.

**3.4.4 Float-16 quantization.** This technique reduces model weights from FP-32 to FP-16, cutting model size in half while maintaining accuracy. However, it does not provide the same performance boost as integer-based quantization.

$$W_{FP16} = \text{cast}(W_{FP32}, FP16) \tag{12}$$

where $W_{FP32}$ and $W_{FP16}$ are the model weights in different formats.

**3.4.5 Weight quantization.** Weight quantization focuses only on reducing model weight precision, keeping activations in FP-32. This approach decreases memory usage while retaining higher accuracy than full quantization.

$$W_q = \text{clip}\left(\text{round}\left(\frac{W}{S}\right), W_{\min}, W_{\max}\right) \tag{13}$$

where $W_{\min}$ and $W_{\max}$ are predefined limits to maintain precision.

**3.4.6 Optimizing model deployment.** After the FastKAN model is converted to TensorFlow Lite format, it is further transformed for deployment on microcontroller-based systems, especially into a .cc array format [36]. This phase is essential for incorporating the model within the limited computing resources typical of TinyML environments. The conversion utilizes a TensorFlow utility script that transforms the *.tflite* file into a *.cc* file including an

unsigned character array (*uint8_t*). This format is crucial for integrating the model directly into the firmware of microcontrollers, enabling efficient storage in the device's read-only memory (ROM).

## 4 Experimental analysis and evaluation

The development, training, and optimization of the Driver Drowsiness Detection model used a mix of cloud and local resources. Model prototyping and training were run in Google Colaboratory with access to a single NVIDIA Tesla V100 GPU (16 GB HBM2, 5120 CUDA cores). Data preparation and supporting experiments were carried out on a local workstation. Additionally, its integration with Google Drive facilitated streamlined data management and experiment monitoring. All implementation was conducted in Python 3.9 to ensure compatibility with widely used machine learning and TinyML frameworks.

Complementary experimental procedures, such as dataset preprocessing, model quantization, and TinyML deployment readiness, were performed locally on a Lenovo Legion 5 15ITH6H laptop. This machine is powered by an 11th Generation Intel(R) Core(TM) i7-11800H processor (2.30 GHz), 16 GB of RAM running at 3200 MHz, and a 6 GB dedicated GPU, operating on a 64-bit Windows 11 Home system. The local setup was adequate for lightweight training, real-time inference validation, and evaluating performance on embedded microcontrollers.

This dual-environment setup enabled a robust and adaptable experimentation workflow, supporting comprehensive development, fine-tuning, and validation of fingerprint authentication models designed for edge-AI and TinyML deployment.

This section provides a thorough evaluation of the FastKAN-based DDD models, emphasizing their classification accuracy, computational efficiency, and suitability for deployment in TinyML environments. The analysis begins by outlining the evaluation metrics. Following this, the training behavior is analyzed across different input image resolutions. The section concludes with an in-depth presentation of test results and a comparison with previous studies.

### 4.1 Evaluation metrics

During the training and testing stages, we evaluated the performances of the ML and TinyML models using commonly employed metrics. These metrics include confusion matrices, from which several evaluation measures are derived, such as accuracy, precision, recall, and F1-score. These metrics assess a model's ability to differentiate between drowsy and awake states, where higher scores indicate superior discrimination capability. Additionally, we considered TinyML-specific metrics such as inference time and model size. The evaluation metrics are presented in Eqs. (14) to (18).

$$\text{Accuracy} = \frac{TP + TN}{TP + TN + FP + FN} \tag{14}$$

$$\text{Precision} = \frac{TP}{TP + FP} \tag{15}$$

$$\text{Recall} = \frac{TP}{TP + FN} \tag{16}$$

$$\text{F1-score} = 2 \times \frac{P \times R}{P + R} \tag{17}$$

$$\text{Inference Time} = \frac{\text{Total Processing Time}}{\text{Number of Inferences}} \tag{18}$$

where TP represents true positive samples, TN denotes true negative samples, FP corresponds to false positive samples, and FN indicates false negative samples.

For each reported performance metric we now provide uncertainty estimates. For test-set accuracy, we report a binomial 95% confidence interval using the Wilson method based on the number of test frames ($N$=3,366). For precision, recall, and F1, we report 95% confidence intervals obtained via a *subject-stratified* bootstrap (2,000 resamples) to respect within-subject correlation; the standard error (SE) is the bootstrap standard deviation. For the primary 64×64 model, test accuracy is 99.94% with 95% CI [99.78, 99.98] and SE $\approx$ 0.04%.

## 4.2 Training and validation process

In order to assess the learning dynamics and generalization performance of the proposed model across different spatial resolutions, a series of training experiments were carried out using varying input image sizes. This section outlines the findings from these experiments, emphasizing the impact of input resolution on model stability, convergence behavior, and its applicability for deployment in environments with limited computational resources.

We initialize all dense layers with Xavier/Glorot and train under fixed conditions (Adam optimizer with learning rate $10^{-3}$, batch size 256, 20 epochs). Across input resolutions, the training/validation curves exhibit smooth convergence and closely aligned trajectories, and the test metrics vary only marginally, indicating that performance is not materially sensitive to the specific random initialization under our protocol. These observations, together with the stable behavior seen across input sizes, suggest that initialization/starting conditions do not drive the reported gains.

To quantify variability and guard against split-specific effects, we conducted subject-disjoint $K$-fold cross-validation on the training pool (48 subjects). We used $K$=5 folds formed at the *subject* level to prevent identity leakage; in each fold, 4 folds were used for training and 1 for validation, following the same training protocol and hyperparameters as the main experiments. We report, for accuracy/precision/recall/F1, the mean $\pm$ standard error (SE) across folds and 95% confidence intervals (fold-wise, subject-stratified bootstrap). The held-out test set (12 unseen subjects) remained untouched and is reported separately. Cross-validated means closely match the single-split results, indicating that the model trained on ~16k samples generalizes well and that performance is not driven by a particular split.

Fig 2 presents the training and validation accuracy and loss trajectories for four different input image resolutions: $150 \times 150$, $128 \times 128$, $64 \times 64$, and $32 \times 32$. These plots provide a comprehensive view of the model's learning dynamics at varying resolutions, shedding light on its ability to generalize while minimizing loss. Models trained on higher resolution inputs ($150 \times 150$ and $128 \times 128$) (see Figs 2a and 2b) exhibit smoother and more stable learning curves with minimal signs of overfitting, as indicated by the close alignment between training and validation accuracy. In contrast, models utilizing lower resolution inputs ($64 \times 64$ and $32 \times 32$) (see Figs 2c and 2d) show greater fluctuations in validation accuracy, suggesting potential challenges in feature extraction due to reduced spatial detail. Nevertheless, the loss curves across all resolutions reveal consistent convergence, affirming the model's capability to effectively optimize its parameters.

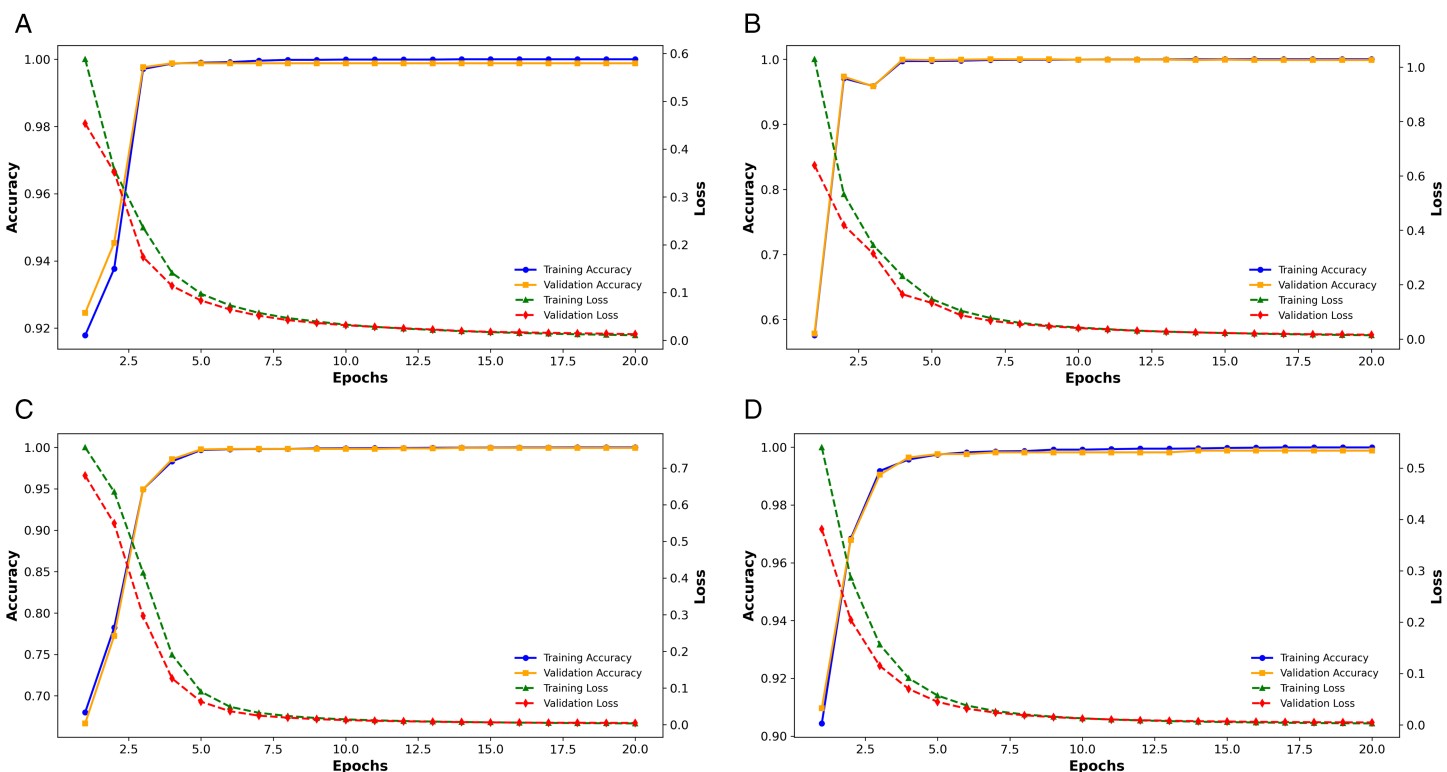

**Fig 2. Training, validation accuracy, and loss curves for different input image sizes.**

Notably, while higher resolution inputs contribute to more stable learning and marginally improved accuracy, lower resolution images offer significant advantages in computational efficiency. This makes them particularly suitable for real-time applications in TinyML environments. The model trained with 32 × 32 resolution achieves competitive performance with notably lower computational overhead, positioning it as a viable solution for deployment in resource-constrained settings.

Table 4 presents a comparative evaluation of the model's performance across different input image resolutions, focusing on key metrics including the total number of trainable parameters, training accuracy, validation accuracy, and test accuracy. The results indicate that higher-resolution inputs (150×150 and 128×128) require substantially more trainable parameters—382,554 and 278,582, respectively. Nevertheless, these configurations achieve near-perfect training and validation accuracy, reflecting strong feature extraction capabilities and effective generalization. Interestingly, even with a notable reduction in parameter count, models trained on lower-resolution inputs (64×64 and 32×32) maintain competitive performance, recording test accuracy values of 99.91% and 99.94%, respectively.

## 4.3 Test process

Fig 3 illustrates the confusion matrices corresponding to various input image resolutions on the test dataset, providing a detailed evaluation of the model's classification performance. Each matrix captures the number of true positives (TP), true negatives (TN), false positives (FP), and false negatives (FN) within a binary classification framework, where class '0' denotes awake states and class '1' represents drowsiness. The model consistently demonstrates

**Table 4. Performance comparison for different input image sizes, including total trainable parameters, training accuracy, validation accuracy, and test accuracy.**

| Image Size | Total Parameters | Training Accuracy (%) | Validation Accuracy (%) | Test Accuracy (%) |
|---|---|---|---|---|
| 150 × 150 | 382,554 | 100.00 | 99.94 | 99.97 |
| 128 × 128 | 278,582 | 99.99 | 99.88 | 99.88 |
| 64 × 64 | 69,686 | 100.00 | 99.94 | 99.91 |
| 32 × 32 | 17,462 | 99.99 | 99.94 | 99.94 |

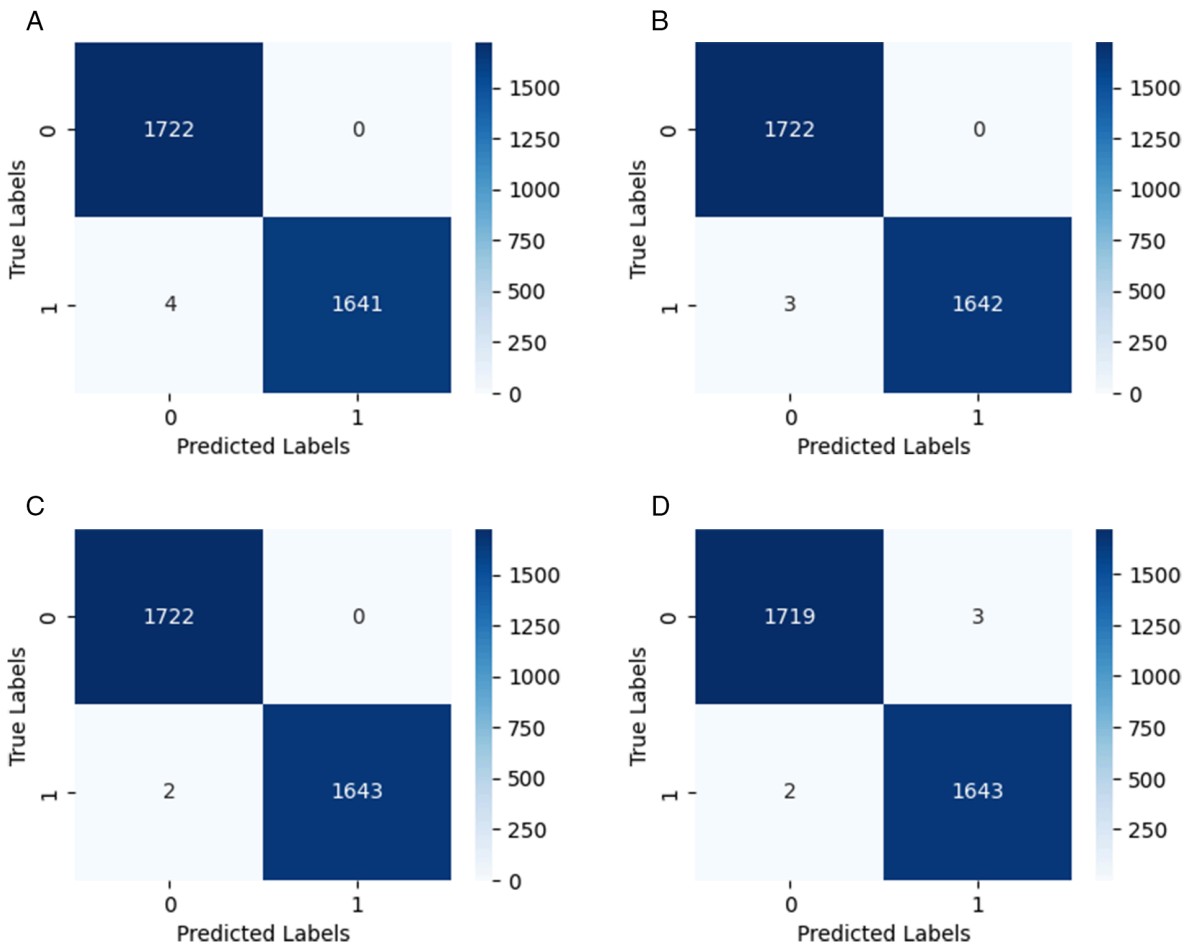

**Fig 3. Confusion matrices for different input image sizes on the test set.**

high classification accuracy across all resolutions, with the majority of predictions correctly aligning with the true labels. Specifically, at a resolution of 150×150, the model successfully identifies 1,722 awake and 1,644 drowsy instances, with only a single misclassification within the drowsy category.

This strong performance trend continues across other resolutions, albeit with slight differences in misclassification rates. The 128×128 resolution exhibits a marginal increase in false positives, with four drowsy instances incorrectly classified as awake. Nevertheless, models trained on 64×64 and 32×32 images maintain commendable performance, registering only three and two false positives, respectively. These results suggest that reducing input resolution

does not substantially impair the model's ability to distinguish between awake and drowsy states.

Table 5 provides a comparative analysis of the FastKAN model's performance across different input image resolutions. Alongside test accuracy, the table includes precision, recall, and F1 score, offering a more holistic evaluation of the model's classification capabilities. These additional metrics closely align with the test accuracy results, affirming the model's consistency and dependability.

Notably, the 64×64 input configuration yields the highest test accuracy of 99.94%, with precision, recall, and F1 score each also reaching 99.94%, indicating exceptional performance. The 128×128 resolution follows closely, achieving a test accuracy of 99.91% with identical scores across the other metrics. Even at the lowest resolution of 32×32, the model maintains impressive effectiveness, attaining a test accuracy of 98.85% and high complementary metrics of 99.85%.

Cross-validated performance across input resolutions is summarized in Table 6; values are means over five subject-disjoint folds.

## 4.4 Conversion and deployment

Table 7 provides a comparative analysis of model size, accuracy, and inference time across various image sizes and quantization techniques. It assesses the influence of post-training quantization (PTQ) on model efficiency and performance, focusing on four distinct image resolutions (150×150, 128×128, 64×64, and 32×32). Multiple quantization strategies, including dynamic quantization, full integer quantization, Float-16 quantization, and weight-only quantization, are evaluated against the original Keras model and the converted .cc array format for embedded deployment. The Keras model serves as the unoptimized baseline, while PTQ methods aim to reduce model size and inference time while maintaining high accuracy.

A notable observation is the significant reduction in model size attained by quantization methods. The PTQ Dynamic, PTQ Float-16, and PTQ Weight-Only models typically attain a 3× to 5× decrease in size relative to their respective Keras models across all picture dimensions. For example, at a resolution of 150×150, the model size diminishes from 1.5 MB (Keras) to 518 KB (PTQ Dynamic), whilst at a resolution of 32×32, the decrease goes from 99 KB to just 35 KB. The .cc array format yields greater file sizes than the quantized models,

**Table 5. Performance metrics for different model versions on the held-out test set (12 unseen subjects).** Values are mean ± standard error (SE), in percentage.

| Sl. No. | Model Version (Dimension) | Test Accuracy (%) | Precision (%) | Recall (%) | F1 Score (%) |
|---|---|---|---|---|---|
| 1 | FastKAN (150 × 150) | 98.88 ± 0.18 | 99.88 ± 0.09 | 99.88 ± 0.09 | 99.88 ± 0.06 |
| 2 | FastKAN (128 × 128) | 99.91 ± 0.05 | 99.91 ± 0.07 | 99.91 ± 0.07 | 99.91 ± 0.05 |
| 3 | FastKAN (64 × 64) | 99.94 ± 0.04 | 99.94 ± 0.06 | 99.94 ± 0.06 | 99.94 ± 0.04 |
| 4 | FastKAN (32 × 32) | 98.85 ± 0.18 | 99.85 ± 0.10 | 99.85 ± 0.10 | 99.85 ± 0.07 |

**Table 6. Subject-disjoint $K$ = 5 cross-validation (training pool: 48 subjects).** Cells report the mean across folds.

| Model (input) | Accuracy (%) | Precision (%) | Recall (%) | F1 (%) |
|---|---|---|---|---|
| 150×150 | 99.97 | 100.00 | 99.94 | 99.97 |
| 128×128 | 99.88 | 100.00 | 99.76 | 99.88 |
| **64×64 (Primary)** | **99.91** | **99.82** | **100.00** | **99.91** |
| 32×32 | 99.94 | 99.88 | 100.00 | 99.94 |

**Table 7. Performance comparison of models with different input image sizes and quantization techniques in terms of model size, accuracy, and inference time.**

| Image Size | Model Type | Model Size | Accuracy (%) | Inference Time (ms) |
|---|---|---|---|---|
| 150 × 150 | Keras (Baseline) | 1.5 MB | – | – |
| | PTQ - Dynamic | 518 KB | 99.88 | 0.75 |
| | PTQ - Full Integer | 389 KB | 51.14 | 1.18 |
| | PTQ - Float16 | 760 KB | 99.88 | 0.85 |
| | PTQ - Weight Only | 518 KB | 99.88 | 0.73 |
| | Converted to .cc Array | 3.1 MB | – | – |
| 128 × 128 | Keras (Baseline) | 1.1 MB | – | – |
| | PTQ - Dynamic | 380 KB | 99.88 | 0.51 |
| | PTQ - Full Integer | 288 KB | 48.86 | 0.71 |
| | PTQ - Float16 | 557 KB | 99.88 | 0.62 |
| | PTQ - Weight Only | 380 KB | 99.88 | 0.57 |
| | Converted to .cc Array | 2.4 MB | – | – |
| 64 × 64 | Keras (Baseline) | 303 KB | – | – |
| | PTQ - Dynamic | 104 KB | 99.94 | 0.14 |
| | PTQ - Full Integer | 84 KB | 48.86 | 0.21 |
| | PTQ - Float16 | 149 KB | 99.94 | 0.16 |
| | PTQ - Weight Only | 104 KB | 99.94 | 0.14 |
| | Converted to .cc Array | 661 KB | – | – |
| 32 × 32 | Keras (Baseline) | 99 KB | – | – |
| | PTQ - Dynamic | 35 KB | 99.85 | 0.05 |
| | PTQ - Full Integer | 33 KB | 48.86 | 0.07 |
| | PTQ - Float16 | 47 KB | 99.85 | 0.05 |
| | PTQ - Weight Only | 35 KB | 99.85 | 0.05 |
| | Converted to .cc Array | 224 KB | – | – |

namely 3.1 MB for 150×150 and 224 KB for 32×32, making it less optimal for storage efficiency. The PTQ Full Integer models, despite attaining minimal model sizes, demonstrate a significant accuracy decline to 48.86%, signifying a loss of essential feature information owing to rigorous quantization. This indicates that PTQ Dynamic, PTQ Float-16, and PTQ Weight-Only quantization offer the optimal equilibrium between model size reduction and performance.

Inference time is a critical dfactor for TinyML applications, where real-time processing is crucial. The findings demonstrate that reduced picture dimensions markedly decrease inference time, with the 32×32 PTQ models attaining a minimum of 0.05 ms, in contrast to 0.73 ms for the 150×150 PTQ model. Among quantization methods, PTQ Weight-Only consistently yields the quickest inference times, rendering it optimal for edge AI implementation. The .cc array format, despite being the final deployment model, has larger file sizes and does not inherently offer inference time improvements. Considering these trade-offs, the optimal model for TinyML deployment is the 32×32 PTQ Weight-Only model, which achieves 99.85% accuracy, a minimal model size of 35 KB, and an exceptionally low inference time of 0.05 ms. This design guarantees excellent performance, low latency, and efficient memory use, rendering it suitable for resource-constrained embedded AI systems.

## 4.5 Real-world deployment challenges

Deploying DDD systems in real-world automotive settings requires careful attention to a range of engineering and environmental constraints. These go beyond achieving high model accuracy, encompassing factors such as hardware limitations, inference latency, operational robustness, energy consumption, and data privacy. The following considerations outline the key challenges and requirements for implementing TinyML-based DDD systems:

1. **Hardware Constraints:** TinyML-targeted embedded platforms typically offer limited RAM, flash storage, and processing capabilities. Deploying large-scale models such as YOLOv5 may not be feasible on these devices without substantial optimization through pruning, quantization, or architectural redesign.

2. **Latency Requirements:** Drowsiness detection systems must generate timely predictions to prevent accidents. Achieving inference times below 100 ms is essential for real-time alerts. Our evaluations show that quantized FastKAN and CNN models achieve inference latencies under 10 ms on microcontrollers, demonstrating their suitability for real-time use.

3. **Environmental Robustness:** Models trained on controlled datasets often struggle under varied real-world conditions—such as low lighting, occlusions, and individual differences in driver appearance. Ensuring consistent performance across changes in camera angles, lighting conditions, and visual obstructions (e.g., masks, sunglasses) is essential for reliable deployment.

4. **Energy Efficiency:** Vehicle-based systems often rely on low-power or battery-operated circuits. TinyML models, such as FastKAN operating at a 32×32 input resolution, exhibit minimal energy consumption per inference, enabling continuous monitoring without compromising battery life.

5. **Data Privacy and Security:** On-device processing of biometric data preserves user privacy by eliminating the need to transmit sensitive information to external servers. Deploying models on microcontroller platforms such as the Arduino Nano and ESP32 supports privacy-by-design compliance and enhances data security.

## 5 Discussion

Table 8 presents a comparative summary of recent research efforts applying TinyML for real-time driver drowsiness detection. In 2023, Norah N. Alajlan et al. [25] reported the highest recorded accuracy of 99.64% using a highly compact CNN model measuring just 0.05 MB, making it highly suitable for deployment on ultra-constrained microcontrollers. In 2024, Sebastian Frey et al. [27] introduced a lightweight CNN with a model size of 21.1 KB that achieved strong performance (91.1% accuracy) on a hybrid EEG+PPG dataset, optimized for the GAP9 platform. Our FastKAN model, trained at an input resolution of 64×64, surpassed all prior models in accuracy, reaching 99.94% while retaining a compact size of only 104 KB. This highlights FastKAN's ability to compete with traditional CNNs in detection accuracy while offering a superior balance of model size, precision, and computational efficiency. Collectively, these findings validate FastKAN's potential for real-time TinyML-based drowsiness detection, combining high robustness with practical deployability for edge-level automotive use cases.

All FastKAN variants in Tables 5 and 7 were trained and evaluated on the same RLDD dataset using the subject-disjoint protocol described in Sect 3.2; the reported metrics are

**Table 8**. Comparison of studies utilizing TinyML for drowsiness detection.

| Ref | Year | Method | Model Size | Accuracy (%) |
|---|---|---|---|---|
| [25] | 2023 | CNN | 0.05 MB | 99.64 |
| [26] | 2023 | CNN | Small | - |
| [27] | 2024 | CNN | 21.1 KB | 91.1 |
| **Proposed Model** | 2025 | FastKAN | 104 KB | 99.94 |

therefore directly comparable across input resolutions and quantization settings. By contrast, the prior works summarized in Table 8 are shown with the metrics reported by their authors on their *original* datasets, and thus serve as a qualitative reference rather than a strictly like-for-like benchmark. Model sizes in Table 7 correspond to the specific artifact indicated: "Keras (Baseline)" denotes the FP32 Keras file size, each "PTQ" entry reports the TensorFlow Lite file size *after* quantization, and "Converted to .cc Array" is the *post-conversion* C array used for microcontroller deployment—which is larger than the TFLite file.

In practical automotive settings, TinyML-based drowsiness detection can be integrated in two main ways: (i) as an OEM driver monitoring feature running on an automotive-grade SoC/MCU connected to an in-cabin RGB/NIR camera, or (ii) as an aftermarket add-on embedded in a dashcam or telematics unit with constrained compute and power budgets. These scenarios impose different design envelopes: OEM deployments typically allow slightly higher input resolutions, more generous SRAM/flash, and tighter integration with vehicle networks (e.g., CAN) for alerting, whereas aftermarket modules prioritize ultra-low power and minimal memory.

Within these envelopes, there is an inherent trade-off between model size, latency, and energy. Smaller, quantized models (e.g., our 64×64, 104 KB FastKAN) achieve sub-10 ms inference on MCUs and reduce energy per decision, but may be more sensitive to domain shift and calibration drift. Conversely, larger input resolutions or de-quantized variants can improve robustness at the cost of latency and power. Sensor modality also plays a role: most public datasets (including UTA-RLDD) use RGB under heterogeneous lighting, whereas many production DMS cameras are NIR, creating a potential RGB→NIR domain gap that must be mitigated (e.g., via augmentation or fine-tuning).

Other real-world limitations include occlusions (glasses, masks), head pose changes due to seat/steering adjustments, and distributional shifts across demographics and environments (night driving, glare). From a systems perspective, techniques such as frame-rate throttling, event-triggered inference, and duty-cycling can materially reduce energy usage without sacrificing safety-critical responsiveness.

Finally, automotive integration requires attention to functional safety processes and software lifecycle discipline, including requirements traceability, validation on in-fleet data, calibration monitoring, and over-the-air update strategies, as well as privacy-by-design via on-device inference. These considerations inform concrete operating points for deployment: when memory and power are most constrained, use INT8-quantized FastKAN at 64×64 with threshold calibration and duty-cycling; where resources permit, modestly higher resolutions or hybrid quantization can increase robustness while maintaining real-time performance.

TinyML deployment imposes strict constraints that demand lightweight yet expressive architectures. In this context, KANs offer significant advantages over conventional MLPs, including greater expressiveness, improved interpretability, and compatibility with aggressive quantization. Table 9 provides a comparative summary of these distinctions, illustrating why FastKAN is particularly well-suited for resource-constrained environments.

## 6 Conclusion and future work

This paper introduced FastKAN-DDD, an innovative driver drowsiness detection model built upon the FastKAN architecture, specifically tailored for TinyML applications.

The proposed model leverages the Kolmogorov–Arnold representation theorem by employing learnable nonlinear activation functions along the network's edges via RBFs. This design enables a compact yet expressive and interpretable model representation. Extensive experimentation using the UTA-RLDD dataset demonstrated that FastKAN-DDD delivers

**Table 9**. Comparative analysis of MLPs and KANs across architectural and deployment-related features.

| Feature | MLPs | KANs |
|---|---|---|
| Activation Functions | ✗ Fixed (e.g., ReLU, Sigmoid) | ✓ Learnable (e.g., spline, RBF) |
| Edge Representation | ✗ Static linear weights | ✓ Nonlinear functional mappings |
| Model Interpretability | ✗ Low (black-box behavior) | ✓ High (functions on edges are visible) |
| Model Compactness | ✗ Often large for high accuracy | ✓ More compact with competitive accuracy |
| Training Dynamics | ✗ Slower convergence | ✓ Faster convergence due to higher expressiveness |
| Quantization Compatibility | ✗ Moderate (loss of precision likely) | ✓ High (well-suited for PTQ) |
| TinyML Deployment | ✗ Requires pruning or compression | ✓ Naturally lightweight and deployable |

high classification accuracy (up to 99.94%), while maintaining a minimal memory footprint (as low as 35 KB) and ultra-fast inference times (down to 0.04 ms), thereby confirming its viability for deployment on computationally constrained microcontrollers. Additionally, post-training quantization techniques were employed to further adapt the model for real-time embedded applications without significant loss in performance.

Despite these encouraging outcomes, the study has some limitations. The system was evaluated using a dataset collected under semi-controlled conditions, which may not fully encompass the complexities and variability of real-world driving scenarios, such as dynamic lighting, occlusions, or diverse driver behaviors. Furthermore, although FastKAN enhances interpretability and model compactness, its performance with multimodal inputs—such as the integration of facial features with EEG or vehicular telemetry data—remains unexplored.

Future research will pursue several avenues, including the expansion of the dataset to encompass a broader spectrum of driving environments, the integration of sensor fusion techniques to address occlusions and suboptimal lighting conditions, and the deployment of the model in real vehicular platforms for extended field testing. Moreover, incorporating edge-to-cloud communication to support federated learning and remote model updates could pave the way for adaptive, scalable, and privacy-aware drowsiness detection solutions.

## Acknowledgment

The authors would like to thank Prince Sultan University for their support.

## Author contributions

**Conceptualization:** Siham Essahraui, Ismail Lamaakal.

**Data curation:** Siham Essahraui, Ismail Lamaakal.

**Formal analysis:** Siham Essahraui, Ismail Lamaakal, Yassine Maleh.

**Funding acquisition:** Hela Elmannai.

**Investigation:** Siham Essahraui, Yassine Maleh.

**Methodology:** Siham Essahraui, Ismail Lamaakal, Yassine Maleh, Khalid El Makkaoui, Mouncef Filali Bouami, Ibrahim Ouahbi, Ahmed A. Abd El-Latif.

**Project administration:** Yassine Maleh, Mouncef Filali Bouami, Hela Elmannai, Ahmed A. Abd El-Latif.

**Resources:** Ismail Lamaakal, Hela Elmannai, Ahmed A. Abd El-Latif.

**Supervision:** Yassine Maleh, Khalid El Makkaoui, Mouncef Filali Bouami, Ibrahim Ouahbi, Hela Elmannai, Ahmed A. Abd El-Latif.

**Validation:** Yassine Maleh, Khalid El Makkaoui, Hela Elmannai.

**Visualization:** Ismail Lamaakal, Yassine Maleh, Khalid El Makkaoui, Mouncef Filali Bouami, Ibrahim Ouahbi.

**Writing – original draft:** Siham Essahraui, Ismail Lamaakal.

**Writing – review & editing:** Yassine Maleh, Khalid El Makkaoui, Mouncef Filali Bouami, Ibrahim Ouahbi, Hela Elmannai, Ahmed A. Abd El-Latif.

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
