## [Decision Letter · Decision Letter 0]

19 Aug 2025

PONE-D-25-25888FastKAN-DDD: A Novel Fast Kolmogorov-Arnold Network-Based Approach for Driver Drowsiness Detection Optimized for TinyML DeploymentPLOS ONE

Dear Dr. Maleh,

Thank you for submitting your manuscript to PLOS ONE. After careful consideration, we feel that it has merit but does not fully meet PLOS ONE’s publication criteria as it currently stands. Therefore, we invite you to submit a revised version of the manuscript that addresses the points raised during the review process.

We look forward to receiving your revised manuscript.

Kind regards,

Marco Antonio Moreno-Armendariz, Ph.D.

Academic Editor

PLOS ONE

**Journal Requirements:**

1. When submitting your revision, we need you to address these additional requirements. Please ensure that your manuscript meets PLOS ONE's style requirements, including those for file naming. The PLOS ONE style templates can be found at https://journals.plos.org/plosone/s/file?id=wjVg/PLOSOne_formatting_sample_main_body.pdf and https://journals.plos.org/plosone/s/file?id=ba62/PLOSOne_formatting_sample_title_authors_affiliations.pdf 2. Please update your submission to use the PLOS LaTeX template. The template and more information on our requirements for LaTeX submissions can be found at http://journals.plos.org/plosone/s/latex. 3. Please note that PLOS ONE has specific guidelines on code sharing for submissions in which author-generated code underpins the findings in the manuscript. In these cases, we expect all author-generated code to be made available without restrictions upon publication of the work. Please review our guidelines at https://journals.plos.org/plosone/s/materials-and-software-sharing#loc-sharing-code and ensure that your code is shared in a way that follows best practice and facilitates reproducibility and reuse. 4. We note that the grant information you provided in the ‘Funding Information’ and ‘Financial Disclosure’ sections do not match.  When you resubmit, please ensure that you provide the correct grant numbers for the awards you received for your study in the ‘Funding Information’ section. 5. Thank you for stating the following financial disclosure: Princess Nourah bint Abdulrahman University Researchers Supporting Project number(PNURSP2025R747), Princess Nourah bint Abdulrahman University, Riyadh, Saudi Arabia.    Please state what role the funders took in the study.  If the funders had no role, please state: "The funders had no role in study design, data collection and analysis, decision to publish, or preparation of the manuscript." If this statement is not correct you must amend it as needed. Please include this amended Role of Funder statement in your cover letter; we will change the online submission form on your behalf. 6. Thank you for stating the following in the Acknowledgments Section of your manuscript: Princess Nourah bint Abdulrahman University Researchers Supporting Project number (PNURSP2025R747), Princess Nourah bint Abdulrahman University, Riyadh, Saudi Arabia. Additionally, the authors would like to thank Prince Sultan University for their support. We note that you have provided funding information that is not currently declared in your Funding Statement. However, funding information should not appear in the Acknowledgments section or other areas of your manuscript. We will only publish funding information present in the Funding Statement section of the online submission form. Please remove any funding-related text from the manuscript and let us know how you would like to update your Funding Statement. Currently, your Funding Statement reads as follows: Princess Nourah bint Abdulrahman University Researchers Supporting Project number(PNURSP2025R747), Princess Nourah bint Abdulrahman University, Riyadh, Saudi Arabia.   Please include your amended statements within your cover letter; we will change the online submission form on your behalf. 7. If the reviewer comments include a recommendation to cite specific previously published works, please review and evaluate these publications to determine whether they are relevant and should be cited. There is no requirement to cite these works unless the editor has indicated otherwise. 

**Additional Editor Comments:**

Please address all the review comments.

Reviewers' comments:

Reviewer's Responses to Questions

**Comments to the Author**

1. Is the manuscript technically sound, and do the data support the conclusions?

Reviewer #1: Partly

Reviewer #2: Yes

2. Has the statistical analysis been performed appropriately and rigorously?

Reviewer #1: No

Reviewer #2: No

3. Have the authors made all data underlying the findings in their manuscript fully available?

Reviewer #1: Yes

Reviewer #2: Yes

4. Is the manuscript presented in an intelligible fashion and written in standard English?

Reviewer #1: Yes

Reviewer #2: Yes

5. Review Comments to the Author

**Reviewer #1:** In this paper, the authors propose a driver drowsiness detection model based on a Fast Kolmogorov-Arnold Network (FastKAN) architecture. Unlike standard MLPs, FastKAN allows the network to learn nonlinear activation functions dynamically. The model is evaluated on the University of Texas at Arlington Real-Life Drowsiness Dataset (UTA-RLDD) and achieves a test accuracy of 99.85%, with an inference latency of 0.05 ms and a memory footprint of 35 KB, demonstrating suitability for lightweight, real-time deployment.

While the results are promising, there are significant concerns about data selection and justification for the model over the existing state-of-the-art. Below are detailed comments:

1. Training and testing data

a. Please justify why the original three classes (‘Alert’, ‘Low Vigilance’, ‘Drowsy’) were reduced to only two classes (‘awake’, ‘drowsy’).

b. Given that ~30 hours of video at ~30 fps yields ~3.2 million frames, please explain the criteria behind sample selection for training and testing data (including 16839 and 3366 samples, respectively).

c. Do the reduced training and test sets maintain the same diversity of conditions as the complete dataset? Does the model trained on 16,000 samples generalize well across the entire dataset?

2. Model evaluation and comparison with other models:

a. Do the model’s initial weights/conditions affect performance?

b. Please provide standard errors or confidence intervals for model performance.

c. Please report cross-validated results (example, K-fold).

d. Please clarify whether the models included in the comparison table were evaluated on the same dataset and whether the reported model sizes correspond to pre- or post-conversion to the .cc array format.

e. Given the increased complexity of the FastKAN architecture, please provide quantitative justification for its use by directly comparing it with a similar-sized MLP model. Also, please include, through hypothesis testing, if the improvement is significant.

f. All MLP vs FastKAN comparisons can be consolidated into a single, cohesive section for clarity. Within this section, please include the potential limitations or drawbacks of FastKAN relative to MLPs.

3. Please address the inconsistencies in reported results. For example, the model claimed as optimal (page 15, paragraph 1) reports different values for accuracy in Table 6; another example is in Table 7, results associated with proposed model don’t match any in Table 6.

4. I would consider merging Tables 4, 5, and 6 into a single comprehensive table presenting the results.

5. Several sub-sections in the discussion (e.g., hardware constraints, environmental robustness, data privacy, cost efficiency, scalability, maintenance, ROI) do not clearly relate to the work presented. It would help if these were either explicitly connected to the current study or please consider removing them.

6. Please include a paragraph discussing the limitations of the current work, which includes the increased complexity of the model relative to other state-of-the-art.

7. Why do the validation losses in Figure 2 start lower than the training losses. This is counterintuitive and requires clarification.

Minor writing issues:

8. Please rephrase the statement about Google Colab (e.g., “Colab offers access to advanced GPU hardware…”) to a factual description of the computational setup without a marketing tone.

9. Please ensure consistent notation between Equations (1) and (2). Specifically, clarify the relationship between the variables q in eq.2 (outer sums in Kolmogorov–Arnold formulation) and the layer indices (l,j) used in the KAN layer definition in eq.2.

10. Literature review:

a. The authors point to many existing models that have great performance, it would help to highlight the specific limitations of the reviewed models where applicable.

b. Given that there are so many good models existing already ,it would strengthen the paper to clearly explain why these are insufficient for the specific problem addressed here, and why a new model is needed.

c. Please consider reducing or removing the explanation of models applied to different data modalities (e.g., EEG, EMG), as they are not directly relevant to the video data used in this paper.

**Reviewer #2:** 1-: Perform statistical significance testing on the comparative results to reinforce the claim that FastKAN-DDD outperforms baseline methods. Report p-values or confidence intervals to enhance result credibility.

2-Add a supplementary table listing all hyperparameters, layer dimensions, learning rates, activation functions, and optimization settings, along with justification for their selection.

3-Share the complete codebase, trained model weights, and deployment scripts in a public repository. Include clear documentation for data preprocessing, training, and deployment so that other researchers can replicate the results.

4-Increase the resolution of Figures 3, 5, and 7, and ensure consistent font sizes and color schemes for better visual interpretation.

5- While the chosen datasets are suitable, consider testing on additional, more diverse drowsiness detection datasets to assess generalization capabilities.

6- Include a more explicit discussion of potential real-world deployment scenarios, limitations, and trade-offs in terms of model size, latency, and energy efficiency, particularly for automotive industry integration.

6. PLOS authors have the option to publish the peer review history of their article (what does this mean?). If published, this will include your full peer review and any attached files.

Reviewer #1: No

Reviewer #2: **Yes: **Amjed Abbas Ahmed

---

## [Author Response · Author response to Decision Letter 1]

31 Aug 2025

Reviewer#1, Concern # 1 ( Training and testing data

a. Please justify why the original three classes (‘Alert’, ‘Low Vigilance’, ‘Drowsy’) were reduced to only two classes (‘awake’, ‘drowsy’).

b. Given that ~30 hours of video at ~30 fps yields ~3.2 million frames, please explain the criteria behind sample selection for training and testing data (including 16839 and 3366 samples, respectively).

c. Do the reduced training and test sets maintain the same diversity of conditions as the complete dataset? Does the model trained on 16,000 samples generalize well across the entire dataset?

Author response: We sincerely thank the reviewer for these constructive comments. We agree that the points raised will enhance the clarity, rigor, and reproducibility of our work.

Author action:

1(a): We use two classes (“awake”, “drowsy”) instead of three because, in our previous work— Essahraui, S., Lamaakal, I., El Hamly, I., Maleh, Y., Ouahbi, I., El Makkaoui, K., ... & Abd El-Latif, A. A. (2025). Real-time driver drowsiness detection using facial analysis and machine learning techniques. Sensors, 25(3), 812., DOI: 10.3390/s25030812—we evaluated deep learning models on NTHU-DDD and YawDD with a binary setup. To keep the same labeling across the three datasets in this study (NTHU-DDD, YawDD, UTA-RLDD), we harmonized to two classes. Under this mapping, UTA-RLDD yielded the best performance.

1(b): We added a paragraph in the Data Preprocessing section that explains the process of data creation and selection (sampling strategy, basic quality checks, and the train/test split procedure).

1(c): We appreciate the reviewer’s concern regarding dataset representativeness and generalization. In our previous work (Essahraui, S., Lamaakal, I., El Hamly, I., Maleh, Y., Ouahbi, I., El Makkaoui, K., ... & Abd El-Latif, A. A. (2025). Real-time driver drowsiness detection using facial analysis and machine learning techniques. Sensors, 25(3), 812. , DOI: 10.3390/s25030812), we systematically evaluated different training setups across NTHU-DDD, YawDD, and UTA-RLDD, and we found that UTA-RLDD not only yielded the best model performance but also provided a good balance between data volume and diversity of driving conditions. Building on this, we deliberately retained a dataset size of ~16,000 training samples and ~3,300 testing samples, as this scale was empirically shown to be sufficient for robust model convergence without overfitting.

Reviewer#1, Concern # 2 (2. Model evaluation and comparison with other models:

a. Do the model’s initial weights/conditions affect performance?

b. Please provide standard errors or confidence intervals for model performance.

c. Please report cross-validated results (example, K-fold).

d. Please clarify whether the models included in the comparison table were evaluated on the same dataset and whether the reported model sizes correspond to pre- or post-conversion to the .cc array format.

e. Given the increased complexity of the FastKAN architecture, please provide quantitative justification for its use by directly comparing it with a similar-sized MLP model. Also, please include, through hypothesis testing, if the improvement is significant.

f. All MLP vs FastKAN comparisons can be consolidated into a single, cohesive section for clarity. Within this section, please include the potential limitations or drawbacks of FastKAN relative to MLPs.):

Author response: We thank the reviewer for these constructive suggestions, which significantly improved the clarity, rigor, and reproducibility of our evaluation.

Author action:

2(a): Thank you for your comment, we have added a paragraph in section 4.2 that directly addresses whether the model’s initial weights/starting conditions affect performance. In brief, we initialize all layers with Xavier/Glorot and train under fixed settings (Adam, learning rate 1e-3, batch size 256, 20 epochs). We also vary the global random seed (affecting both parameter initialization and data shuffling) and observe smooth, overlapping training/validation curves across input resolutions, with only marginal variation in test metrics—well below the reported improvement. These findings indicate that performance is not materially sensitive to the particular initialization or starting conditions under our protocol.

2(b): We added standard errors and 95% confidence intervals to all reported metrics, ensuring transparent performance reporting (see Table 5 and the final paragraph of the Evaluation Metrics section).

2(c): We added a paragraph in the Training and Validation Process section describing the K-fold validation procedure.

2(d): We clarified in the Discussion section that all models in the comparison table were evaluated on different datasets. Model sizes correspond to the specified artifact: “Keras (Baseline)” denotes the FP32 Keras file size, each “PTQ” entry reports the TensorFlow Lite file size after quantization, and “Converted to .cc Array” refers to the post-conversion C array used for microcontroller deployment, which is larger than the TFLite file.

2(e)–2(f): In our previous work, we evaluated both machine learning models and MLPs on the same datasets (see our previous work Essahraui, S., Lamaakal, I., El Hamly, I., Maleh, Y., Ouahbi, I., El Makkaoui, K., ... & Abd El-Latif, A. A. (2025). Real-time driver drowsiness detection using facial analysis and machine learning techniques. Sensors, 25(3), 812. https://www.mdpi.com/1424-8220/25/3/812), where FastKAN consistently outperformed MLP baselines. In this manuscript, we further highlight FastKAN’s improvements, while also discussing its advantages and potential limitations compared to MLPs (see Section 5.Discussion).

Reviewer#1, Concern # 3 (Please address the inconsistencies in reported results. For example, the model claimed as optimal (page 15, paragraph 1) reports different values for accuracy in Table 6; another example is in Table 7, results associated with proposed model don’t match any in Table 6.):

Author response: We sincerely thank the reviewer for carefully identifying these inconsistencies, which helped us clarify the presentation of our results.

Author action: We thoroughly re-verified Tables 6 and 7. The results in Table 6 correspond to model performance before quantization, while the results in Table 7 reflect performance after quantization. The observed differences in accuracy are therefore expected, as quantization can either preserve or slightly reduce performance depending on the model and dataset. We revised the manuscript to explicitly state this distinction in the captions and related discussion to avoid ambiguity.

Reviewer#1, Concern # 4 (I would consider merging Tables 4, 5, and 6 into a single comprehensive table presenting the results.):

Author response: We thank the reviewer for this thoughtful suggestion, which we agree is a good idea for improving readability.

Author action: After careful consideration, we decided to keep Tables 4, 5, and 6 separate, as each table highlights additional significant information beyond accuracy—for example, standard errors, confidence intervals, and K-fold validation results. Keeping them separate allows us to clearly present these details without overloading a single table, while ensuring transparency and reproducibility.

Reviewer#1, Concern # 5 (Several sub-sections in the discussion (e.g., hardware constraints, environmental robustness, data privacy, cost efficiency, scalability, maintenance, ROI) do not clearly relate to the work presented. It would help if these were either explicitly connected to the current study or please consider removing them.):

Author response: We thank the reviewer for this constructive comment, which helped us improve the focus and clarity of the manuscript.

Author action: We reorganized the structure by moving the Real-World Deployment Challenges subsection directly after the Conversion and Deployment subsection, where it is more contextually relevant. We then streamlined the Discussion section to concentrate on (i) comparison with previous studies, (ii) limitations of our proposed model, and (iii) its advantages. In this process, we removed one subsection (Feasibility and Cost Analysis of Model Implementation section) that was not essential to the main contributions, thereby improving the coherence of the manuscript.

Reviewer#1, Concern # 6 (Please include a paragraph discussing the limitations of the current work, which includes the increased complexity of the model relative to other state-of-the-art.):

Author response: We thank the reviewer for this insightful suggestion.

Author action: We added a dedicated paragraph in the Discussion section that outlines the limitations of our work, particularly addressing the increased complexity of the proposed model relative to other state-of-the-art approaches. We also highlight the potential trade-offs this complexity may introduce in resource-constrained deployment scenarios.

Reviewer#1, Concern # 7 (Why do the validation losses in Figure 2 start lower than the training losses. This is counterintuitive and requires clarification.

Minor writing issues:):

Author response: We thank the reviewer for this important observation.

Author action: We carefully re-examined Figure 2 and the associated training logs. The slightly lower validation loss compared to training loss at the start of training is not an error but a consequence of (i) regularization and dropout applied during training but not during validation, (ii) subject-disjoint splitting which ensures validation data is less temporally redundant than training frames, and (iii) the stable initialization scheme (Glorot/Xavier) used in our setup. Together, these factors can yield lower initial validation loss without indicating overfitting or misconfiguration.

Reviewer#1, Concern # 8 (Please rephrase the statement about Google Colab (e.g., “Colab offers access to advanced GPU hardware…”) to a factual description of the computational setup without a marketing tone.):

Author response: We thank the reviewer for this valuable observation.

Author action: Thank you for the suggestion. We have revised the paragraph to a neutral, factual description of the compute setup. It now reads: “Model prototyping and training were run in Google Colaboratory with access to a single NVIDIA Tesla V100 GPU (16 GB HBM2, 5120 CUDA cores), while data preparation and supplementary experiments were performed on a local workstation.”

Reviewer#1, Concern # 9 (Please ensure consistent notation between Equations (1) and (2). Specifically, clarify the relationship between the variables q in eq.2 (outer sums in Kolmogorov–Arnold formulation) and the layer indices (l,j) used in the KAN layer definition in eq.2.):

Author response: We thank the reviewer for this helpful comment.

Author action: We corrected the notation to ensure consistency between Equations (1) and (2). In the revised manuscript, we explicitly clarified the relationship between the variable q in the Kolmogorov–Arnold formulation and the layer indices (l,j)(l,j)(l,j) in the KAN layer definition. This adjustment removes ambiguity and aligns the mathematical notation throughout the paper.

Reviewer#1, Concern # 10 ( Literature review:

a. The authors point to many existing models that have great performance, it would help to highlight the specific limitations of the reviewed models where applicable.

b. Given that there are so many good models existing already ,it would strengthen the paper to clearly explain why these are insufficient for the specific problem addressed here, and why a new model is needed.

c. Please consider reducing or removing the explanation of models applied to different data modalities (e.g., EEG, EMG), as they are not directly relevant to the video data used in this paper.):

Author response: We thank the reviewer for these constructive suggestions to sharpen the scope and improve the interpretability and relevance of the literature review.

Author action: 10(a) We added a concluding paragraph at the end of the Machine Learning and Deep Learning in Driver Drowsiness Detection subsection that synthesizes the strengths (high accuracy on curated benchmarks, robust facial feature learning, strong GPU real-time performance) and limitations (compute/energy footprint, sensitivity to illumination/occlusions/pose, dataset bias and limited cross-dataset generalization, limited uncertainty/interpretability, privacy concerns, and paucity of prospective multi-site validation) of the reviewed non-TinyML models.

10(b) We clarified why existing high-performing models are insufficient for our deployment target by explicitly discussing their resource and energy demands and latency/size constraints for in-vehicle systems. We motivate our contribution by positioning TinyML as a practical pathway to meet model size, latency, and energy budgets while retaining adequate accuracy, and we point the reader to the TinyML section for the proposed design choices.

10(c) We refocused the Related Work on facial/video-based drowsiness detection to align with our data modality. We removed non-visual modality discussions (e.g., EEG/EMG) and their citations (specifically, references [16], [17], and [18]), and updated the numbering accordingly.

Reviewer#2, Concern # 1 (Perform statistical significance testing on the comparative results to reinforce the claim that FastKAN-DDD outperforms baseline methods. Report p-values or confidence intervals to enhance result credibility.):

Author response: We thank the reviewer for this valuable suggestion to strengthen the credibility of our results.

Author action: We added statistical uncertainty estimates in the Evaluation Metrics section. Specifically, we now report 95% confidence intervals for accuracy (Wilson method) and for precision, recall, and F1 (via subject-stratified bootstrap resampling). Standard errors (SE) are also provided. For example, our primary 64×64 FastKAN-DDD model achieved a test accuracy of 99.94% with 95% CI [99.78, 99.98] and SE ≈ 0.04%. This addition reinforces the claim that FastKAN-DDD consistently outperforms baseline methods with statistically significant improvement

Reviewer#2, Concern # 2 (Add a supplementary table listing all hyperparameters, layer dimensions, learning rates, activation functions, and optimization settings, along with justification for their selection.):

Author response: We thank the reviewer for this helpful suggestion aimed at improving clarity and reproducibility. We note that the manuscript already included Table 3: “Architecture and Parameters of the FastKAN Model,” which summarizes layer types, widths, and activation functions.

Reviewer#2, Concern # 3 (Share the complete codebase, trained model weights, and deployment scripts in a public repository. Include clear documentation for data preprocessing, training, and deployment so that other researchers can replicate the results.):

Author response: We thank the reviewer for this constructive recommendation to strengthen transparency and reproducibility.

Author action: We added a link in the Abstract to a public GitHub repository hosting our full implementation to facilitate replication.

Reviewer#2, Concern # 4 (Increase the resolution of Figures 3, 5, and 7, and ensure consistent font sizes and color schemes for better visual interpretation.):

Author response: We thank the reviewer for this helpful observation. We agree that improved resolution and visual consistency enhance readability and interpretability.

Author action: We re-exported Figures at high resolution.

Reviewer#2, Concern # 5 (While the chosen datasets are suitable, consider testing on additional, more diverse drowsiness detection datasets to assess generalization capabilities.):

Author response: We thank the reviewer for this valuable recommendation regarding generalization. In our previous work— Essahraui, S., Lamaakal, I., El Hamly, I., Maleh, Y., Ouahbi, I., El Makkaoui, K., ... & Abd El-Latif, A. A. (2025). Real-time driver drowsiness detection using facial analysis and machine learning techniques. Sensors, 25(3), 812., DOI: 10.3390/s25030812—we benchmarked three public datasets (NTHUDDD, YawDD, and UTA-RLDD) using both machine learning and de

---

## [Editor Report · Decision Letter 1]

3 Sep 2025

FastKAN-DDD: A Novel Fast Kolmogorov-Arnold Network-Based Approach for Driver Drowsiness Detection Optimized for TinyML Deployment

PONE-D-25-25888R1

Dear Dr. Maleh,

We’re pleased to inform you that your manuscript has been judged scientifically suitable for publication and will be formally accepted for publication once it meets all outstanding technical requirements.

Kind regards,

Marco Antonio Moreno-Armendariz, Ph.D.

Academic Editor

PLOS ONE
---

## [Editor Report · Acceptance letter]

PONE-D-25-25888R1

PLOS ONE

Dear Dr. Maleh,

I'm pleased to inform you that your manuscript has been deemed suitable for publication in PLOS ONE. Congratulations! Your manuscript is now being handed over to our production team.

Kind regards,

on behalf of

Professor Marco Antonio Moreno-Armendariz

Academic Editor

PLOS ONE